# Overexpression of KLF4 Suppresses Pulmonary Fibrosis through the HIF-1α/Endoplasmic Reticulum Stress Signaling Pathway

**DOI:** 10.3390/ijms241814008

**Published:** 2023-09-12

**Authors:** Shanchen Wei, Fei Qi, Yanping Wu, Xinmin Liu

**Affiliations:** 1Department of Geriatric, Peking University First Hospital, Beijing 100034, China; 18811779077@163.com; 2School Of Clinical Medicine, Anhui Medical College, Hefei 230032, China; qifei521@pku.edu.cn; 3Department of Chemotherapy, Peking University First Hospital, Beijing 100034, China

**Keywords:** Krüppel-like factor 4, pulmonary fibrosis, endoplasmic reticulum stress, HIF-1α

## Abstract

The hypoxia-inducible factor-1α/endoplasmic reticulum stress signaling pathway (HIF-1α/ERS) has a crucial role in the pathogenetic mechanism of pulmonary fibrosis (PF). However, the upstream regulatory mediators of this pathway remain unclear. In the present study, by conducting bioinformatics analysis, we found that Krüppel-like factor 4 (KLF4) expression was decreased in the lung tissues of patients with idiopathic pulmonary fibrosis (IPF) as compared to that in patients with non-IPF. Furthermore, KLF4 expression was significantly reduced (*p* = 0.0331) in bleomycin-induced fibrotic HFL-1 cells. Moreover, in mice with bleomycin-induced PF, the degree of fibrosis was significantly reduced in mice overexpressing KLF4 as compared to that in wild-type mice. In mice and HFL-1 cells, KLF4 overexpression significantly reduced bleomycin-induced protein expression of HIF-1α (*p* = 0.0027) and ERS markers, particularly p-IRE1α (*p* = 0.0255) and ATF6 (*p* = 0.0002). By using the JASPAR database, we predicted that KLF4 has five binding sites for the HIF-1α promoter. The results of in vitro and in vivo studies suggest that KLF4 may inhibit PF through the HIF-1α/ERS pathway. This finding could guide the development of future therapies for PF and facilitate the identification of appropriate biomarkers for routine clinical diagnosis of PF.

## 1. Introduction

Idiopathic pulmonary fibrosis (IPF), a chronic terminal lung disease, causes progressive aggravation of dyspnea and eventual respiratory failure, resulting in patient death [1,2,3]. Fibroblast proliferation and extracellular matrix remodeling are the characteristics of IPF, which lead to permanent scarring of the lung [4,5,6].

The endoplasmic reticulum (ER) regulates protein homeostasis and plays an important role in protein folding and processing. Any factor that disrupts the protein folding and processing mechanism results in the accumulation of misfolded proteins in the ER; this phenomenon is known as endoplasmic reticulum stress (ERS). An imbalance between the demand of the cell for protein synthesis and the ER’s ability to process unfolded or damaged proteins stimulates the cell’s own protective pathway, which is known as the unfolded protein response (UPR) that functions to re-establish ER homeostasis [7]. ERS and UPR affect fibroblast differentiation into myofibroblasts, fibroblast activation, macrophage polarization, inflammatory response activation, and other fibrotic processes [8,9,10,11]. Several factors can cause ERS, including gene mutation, infection, inhaled particulate matter such as cigarette smoke, hypoxia, and aging [12]. These factors activate UPR, platelet-derived growth factor, transforming growth factor-β1 (TGF-β1), Wnt, chemokines, and other fibrosis-induced factors and pathways to ultimately induce apoptosis [13].

ERS is closely associated with hypoxia-related pathways. Hypoxia-inducible factor-1α (HIF-1α) induces ERS and promotes apoptosis in alveolar epithelial cells [14]. VEGF, a downstream factor of HIF signaling, regulates the expression of protein kinase RNA-like ER kinase (PERK) and activating transcription factor 6 (ATF6) [15], thus suggesting a regulatory role of HIF signaling for ERS. A recent study revealed that the hypoxia pathway interacts with ERS [16]. However, the upstream regulatory mediators of the HIF-1α/ERS pathway remain unclear.

Krüppel-like factors (KLFs) are zinc finger proteins that mediate gene transcription in eukaryotic cells. KLFs regulate the expression of genes with GC-rich promoters and are involved in the regulation of cell proliferation, aging [17], differentiation [18], embryonic development [19], organ formation [20], tumors infiltration [21], and other biological behaviors [22]. KLF4, a KLF family member, is a pleiotropic transcription factor with context-dependent activation or repression activity. Our previous study revealed that KLF4 attenuates bleomycin (BLM)-induced pulmonary fibrosis (PF) by inhibiting TGF-β1-induced epithelial–mesenchymal transition (EMT) [23]. Overexpression of KLF4 can also inhibit the invasion and migration of non-small cell lung cancer cells through the c-Jun-NH2-terminal kinase/EMT signaling pathway [24].

Thus, KLF4 and HIF1α/ERS play a critical role in the pathogenetic mechanism of PF. However, the effect of KLF4 on the HIF1α/ERS pathway has not yet been investigated. Hence, the present study aimed to determine how KLF4 affects the HIF-1α/ERS pathway by using the bleomycin (BLM)-induced PF mice model and the fibrotic HFL-1 cell line.

## 2. Results

### 2.1. KLF4 Expression Was Decreased in Patients with IPF and in BLM-Induced Fibrotic HFL-1 Cells

The GSE150910 dataset containing the data collated from the tissues of 103 patients with IPF and 103 healthy controls was downloaded from the GEO database and analyzed. A total of 1755 differentially expressed genes (DEGs) (538 downregulated and 1217 upregulated) were identified, and their distribution was shown using a volcano plot (Figure 1a). Patients with IPF showed a reduction in KLF4 expression.

The HFL-1 cell line was used for in vitro experiments. Different concentrations of BLM were used to induce HFL-1 cells. The cell viability was not significantly affected at the BLM concentration of 100–200 µg/mL (Figure 1b). Collagen Ⅰ and collagen Ⅲ mRNA expression levels were significantly increased (Figure 1c,d). At the BLM concentration of 200 ug/mL, the protein expression levels of collagen Ⅰ and collagen Ⅲ were significantly increased after HFL-1 cell induction for 24 h (Figure 1e,f). KLF4 expression was also decreased in BLM-induced cells as compared to that in the control group (Figure 1g).

### 2.2. Overexpression of KLF4 Attenuated BLM-Induced PF in a Mice Model

To study the effect of KLF4 on PF in vivo, transgenic mice overexpressing KLF4 were generated. PF was induced in both KLF4-overexpressing mice and wild-type mice by intratracheal instillation of BLM. Wild-type mice showed the development of severe PF. However, KLF4-overexpressing transgenic mice showed a decrease in the severity of PF and the accumulation of collagen fibers (Figure 2a,b). Moreover, compared to wild-type mice, KLF4-overexpressing mice displayed less damage of the ER and mitochondria (Figure 2c,d).

### 2.3. Overexpression of KLF4 Attenuated the Level of ERS in BLM-Induced PF

To understand how KLF4 affects BLM-induced ER stress, HFL-1 cells were infected with the adenoviral vectors AdKLF4 and Ad tetracycline transactivator (AdtTA) for 24 h and subsequently stimulated with BLM for 0, 3, 6, 9, 12, 15, 18, and 24 h. The ERS levels were altered. Western blotting assay showed that KLF4 overexpression inhibited the expression of p-IRE1α and ATF6; however, BIP expression was not inhibited (Figure 3a). Furthermore, with the increase in induction time, this inhibitory effect remained relatively stable (Figure 3a).

In addition to the in vitro cell line, we also established a mouse model of PF by tracheal instillation of BLM in KLF4-overexpressing FVB mice. RNA was extracted from the collected lung tissues and subjected to quantitative reverse transcription-polymerase chain reaction (qRT-PCR). In the saline group, KLF4-overexpressing mice and wild-type mice showed no significant differences in ER stress levels (Figure 3b); however, in the BLM-induced model group, KLF4 overexpression significantly inhibited the mRNA expression of the ERS markers IRE1α and ATF6 (Figure 3c).

### 2.4. Overexpression of KLF4 Inhibited the Induced Expression of HIF-1α

HFL-1 cells were stimulated with 200 µg/mL BLM. The cytoplasmic and nuclear proteins were collected at different time points. The protein expression level was detected by a Western blotting assay. Following BLM stimulation for 3 h, the cytoplasmic expression of KLF4 and HIF-1α decreased and increased significantly, respectively (Figure 4a). After 6 h, the expression levels of both proteins returned to the baseline level (Figure 4a). In the nucleus, KLF4 expression was significantly decreased at 3 h and then returned to the baseline level, and HIF-1α expression remained at 6 h (Figure 4b).

Total RNA was extracted from mice tissues and subjected to qPCR. In the saline group, KLF4-overexpressing mice and wild-type mice showed no significant differences in HIF-1α expression (Figure 4c); in contrast, in the BLM model group, KLF4 overexpression significantly inhibited the induced expression of HIF-1α (Figure 4d). By using the JASPAR database, four KLF4-binding sites in the DNA promoter region of HIF-1α were predicted (Figure 4e).

## 3. Discussion

Previous studies have shown that KLF4 affects PF by regulating multiple downstream mediators, such as TERT [25], E-cadherin [23], and plasminogen activator inhibitor-1 [26]. Although most studies have confirmed that KLF4 inhibits PF progression, a few studies have reported the opposite effect [27]. In another study, the HIF-1α expression level was positively correlated with fibrosis severity, and the inhibition of HIF-1α alleviated bleomycin-induced PF [28]. Additionally, HIF-1α regulated EMT in paraquat intoxication-induced early lung fibrosis through HIF-1α via the Snail and β-catenin pathways [29]. Hypoxia and HIF-1α can trigger ERS and CHOP-mediated apoptosis in alveolar epithelial cells, thus suggesting their potential contribution to IPF development [30].

In the present study, we demonstrated that KLF4 expression is decreased in IPF. Moreover, KLF4 overexpression inhibits the HIF-1α/ERS pathway and alleviates PF. Based on this, we believe that the expression of KLF4 is decreased in patients with IPF, and its transcriptional suppression on HIF-1α is downregulated. When suffering hypoxia, the expression of HIF-1α in patients with IPF increases much higher than normal, which activates UPR. This affects the differentiation process of fibroblasts into myofibroblasts, fibroblast activation, macrophage polarization, inflammatory response activation, and other fibrotic processes [8,9,10,11]. Ultimately, it leads to the occurrence and progression of pulmonary fibrosis.

The present study has some limitations. The sample size is relatively small, and the statistical results may be biased. We did not examine the specific molecular mechanisms underlying the effect of KLF4 overexpression. The transcription factor KLF4 was predicted to bind to the five sites in the HIF-1α promoter through bioinformatics analysis. In future studies, we will conduct a CHIP assay and an electrophoresis mobility shift assay to confirm the binding of the two proteins and analyze the specific binding sites.

Our present study confirmed that KLF4 may affect the progression of PF by regulating the HIF-1α/ERS pathway; this finding provides a new mechanism for the results of previous research.

APTO-253, a small-molecule inducer of KLF4, was found to restore KLF4 expression in fibrotic fibroblasts and induced remission in an experimental model of clinically relevant persistent PF [31]. Our study provides support for the clinical application of APTO-253 in treating PF. The findings of our study could also serve as support for identifying biomarkers for routine clinical application in PF diagnosis and treatment.

## 4. Materials and Methods

### 4.1. Bioinformatics

The GSE150910 dataset containing the RNA-sequencing data of tissues from patients with IPF and healthy controls was downloaded from the GEO database (https://www.ncbi.nlm.nih.gov/geo/, accessed on 18 October 2022). R software, version R-4.0.5 was then used to screen the DEGs. The criteria for the selection of DEGs were as follows: −log_10_ *p*-value < 0.05 and a |fold change| > 2. The binding sites of KLF4 in the HIF-1α promoter were predicted by the online prediction website, JASPAR 3.0 (http://jaspar.genereg.net, accessed on 15 December 2022).

### 4.2. Generation of KLF4-Overexpressing Mice

Mice overexpressing KLF4 were generated by injecting KLF4 plasmid DNA into the zygote of FVB mice by following the standard pronuclear injection process developed by Cyagen Biosciences (Santa Clara, CA, USA), as described previously in [23]. Positive founders were identified by PCR. Genotyping was conducted by PCR on toe DNA by using the following primers for KLF4: (forward) 5′-CCGATGAACTGACCAGGCACTA-3′ and (reverse) 5′-AGCGAGGAAGCGGAAGAGC-3′. Wild-type littermates were used as controls. No differences in weight or survival rate were observed between the KLF4 overexpression group and the wild-type group. Mice were housed in a temperature- and humidity-controlled specific pathogen-free facility, with standard chow and water provided ad libitum. Mice were maintained on a 12-h light/dark cycle at 22–25 °C with 45–65% humidity. All animal care and experimental procedures conformed to the Guide for the Care and Use of Laboratory Animals (NIH Publication no. 85-23, revised 1996), and were approved by the Animal Research Committee of Peking University First Hospital. All experiments were performed in accordance with relevant guidelines and regulations.

### 4.3. BLM-Induced Fibrosis Model

Male FVB mice with an average weight of approximately 25–30 g and aged 8–10 weeks were given intratracheally instilled saline or 5 mg/kg BLM on day 0, as described previously [23]. Mice were sacrificed on day 21. The experiments were conducted using 6 mice per group. The lung tissues were collected, fixed in formalin, embedded in paraffin, sectioned, and stained with Masson’s trichrome stain or subjected to immunohistochemical analysis.

### 4.4. Transmission Electron Microscopy

Transmission electron microscopy was performed as described previously [32]. Briefly, tissues were fixed with 1.5% glutaraldehyde for 4 h, followed by post-fixing with 1% osmium tetroxide for 2 h. Double-distilled water was used to wash the samples for 5 min each time. The samples were then dehydrated in increasing concentrations of ethanol (50%, 70%, 90%, and 100%) and anhydrous acetone (3 times for 15 min each). The samples were then sequentially saturated with 1:1 and 1:2 mixtures of acetone and embedding medium, respectively. Subsequently, the samples were incubated at 37 °C for 12 h, followed by incubation at 60 °C for 48 h to complete the polymerization procedure. Next, 60-nm ultrathin sections (LEICA EM UC7) stained with 1% uranyl acetate were imaged using a JEM-1400 Flash transmission electron microscope (JEOL, Tokyo, Japan).

### 4.5. Cell Culture

HFL-1 cells were purchased from the National Collection of Authenticated Cell Cultures (Shanghai, China) (https://cellbank.org.cn/orderlist.php, accessed on 24 October 2022, Accession numbers: SCSP-5049) and cultured in Ham’s F-12 (Biological Industries, Beit Haemek, Israel) supplemented with 10% fetal bovine serum (Gibco, Waltham, MA, USA) at 37 °C in 5% CO_2_. Different concentrations of BLM were used to induce the fibrotic phenotype in HFL-1 cells, and the optimal concentration of BLM was 200 µg/mL.

### 4.6. Adenoviral Vectors and Infections

The KLF4 adenovirus was constructed as previously described [24,33]. The expression of the inserted KLF4 plasmid DNA was driven by a 7 × tet operon/minimal cytomegalovirus promoter, which was further controlled by a tetracycline-controlled transactivator (tTA). The adenoviruses were purified by cesium chloride methods. For adenovirus-mediated gene transfer, confluent cell lines were exposed to adenoviral vectors with tetracycline transactivator adenovirus (Ad-tTA) to induce tetracycline-controllable expression. The cells were co-infected with AdKLF4 and AdtTA (20 multiplicity of infection) and incubated for 6 h with or without tetracycline (0.1 μg/mL).

### 4.7. Histology and Immunohistochemistry Analysis

For histological analysis to assess PF, consecutive 5-μm-thick sections were prepared from the lung tissues. The sections were stained with Masson’s trichrome stain and observed under a microscope as described previously [34]. For immunohistochemical analysis of the expression of collagen III (22734-1-AP, Proteintech, Wuhan, China), the sections were stained using the immunohistochemical method. To sum up briefly, the sections were incubated first with primary antibodies overnight at 4 °C, and then with the biotinylated goat anti-rabbit IgG for 30 min at room temperature. An immunohistochemical detection system (EnVision kit, Dako, Kyoto, Japan) was used for visualization.

### 4.8. Western Blotting Assay

Total protein was extracted from mouse tissues and cell lines by using a protein extraction buffer containing a 1% protease inhibitor cocktail (Targetmol) and a phosphatase inhibitor (Cat. No. 4906845001, Roche, Basel, Switzerland). The cell lysate was centrifuged at 4 °C at 14,000 rpm for 15 min. The obtained proteins were separated by 10% sodium dodecyl sulfate-polyacrylamide gel electrophoresis (SDS-PAGE) and transferred to polyvinylidene fluoride membranes (Millipore, Burlington, MA, USA). After blocking with 5% skimmed milk for 1 h at room temperature, the membranes were incubated at 4 °C overnight with the following primary antibodies: KLF4 (ab215036, Abcam, Cambridge, UK), HIF1α (66730-1-Ig, Proteintech), p-IRE1α (AP1146, ABclonal, Wuhan, China), BIP (ab108615, Abcam), ATF6 (ab227830, Abcam), Collagen I (14695-1-AP, Proteintech), Collagen III (22734-1-AP, Proteintech), GAPDH (10494-1-AP, Proteintech), and β-Tubulin (10068-1-AP, Proteintech). After washing with Tris-buffered saline with 1% Tween-20 (TBST), the membranes were incubated with HRP-conjugated secondary antibodies (ZSGB-Bio, Beijing, China) for 1 h at room temperature. The blots were visualized using an enhanced chemiluminescence reagents detection kit (Amersham Biosciences Fairfield, London, UK).

### 4.9. qRT-PCR

Total RNA was extracted from HFL-1 cells or mouse lung tissues by using Trizol reagent (Invitrogen, Carlsbad, CA, USA). Next, 2 μg of total RNA was converted into cDNA by using reverse transcriptase and oligo (dT) (Promega, Madison, MI, USA) as a primer. Real-time quantitative PCR was performed using the iQ™ SYBR Green PCR Supermix in the DNA Engine Opticon real-time system (Bio-Rad Laboratories, Inc., Hercules, CA, USA) with GAPDH as an internal control. The primer sequences were as follows: KLF4, 5′-CCGATGAACTGACCAGGCACTA-3′ (forward), 5′-AGCGAGGAAGCGGAAGAGC-3′ (reverse); GAPDH, 5′-ACCACAGTCCATGCCATCAC-3′ (forward), 5′-TCCACCACCCTGTTGCTGTA-3′ (reverse); HIF-1α, 5′-AGGATGAGTTCTGAACGTCGAAA-3′ (forward), 5′-CTGTCTAGACCACCGGCATC-3′ (reverse); IRE1α, 5′- TGTTTGTCTCGACCCTGGATG-3′ (forward), 5′-CGTTGTTCTTGCCTCCAAGTG-3′ (reverse); BIP, 5′-CCTGCGTCGGTGTGTTCAAG-3′ (forward), 5′-AAGGGTCATTCCAAGTGCG-3′ (reverse); and ATF-6, 5′-CAGCCCCTGTGGTGAGCAGC-3′ (forward), 5′-GCAGCCTTGAGCCTGGCCTC-3′ (reverse).

### 4.10. Statistical Analysis

All statistical analyses were conducted by using the SPSS software, version 24.0. The differences in the qRT-PCR assay results between the two sample groups were analyzed using an independent *t*-test when the data was normally distributed; otherwise, a nonparametric test was used. The Wilcoxon signed-rank test was used for the Western blotting assay. A *p*-value of <0.05 was considered statistically significant.

## Figures and Tables

**Figure 1 ijms-24-14008-f001:**
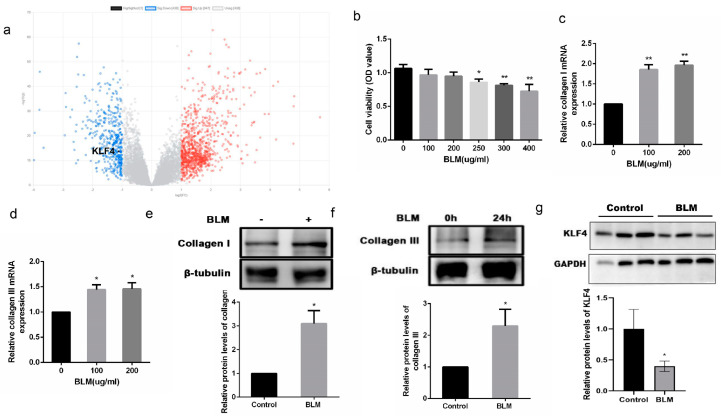
KLF4 expression is downregulated in human IPF lung tissue and in bleomycin-induced fibrosis of HFL-1 cells in vitro. A volcano map exhibits significant number of differentially expressed genes between the IPF and control groups (**a**). Red, blue, and gray bubbles imply upregulated genes, downregulated genes, and nonsignificantly-changed genes, respectively. Different concentrations (0, 100, 200, 250, 300, and 400 µg/mL) of bleomycin were used to induce HFL-1 cells. After 24 h, the cell viability was tested (**b**), and the mRNA expression levels of collagen I and collagen Ⅲ were determined by qRT-PCR (**c**,**d**). At 200 µg/mL BLM concentration, a Western blotting assay was conducted to determine the protein expression levels of collagen Ⅰ, collagen Ⅲ, and KLF4 (**e**–**g**). The values are expressed as mean ± SD. *n* = 3 for each group. * *p* < 0.05 and ** *p* < 0.01 for *t*-test.

**Figure 2 ijms-24-14008-f002:**
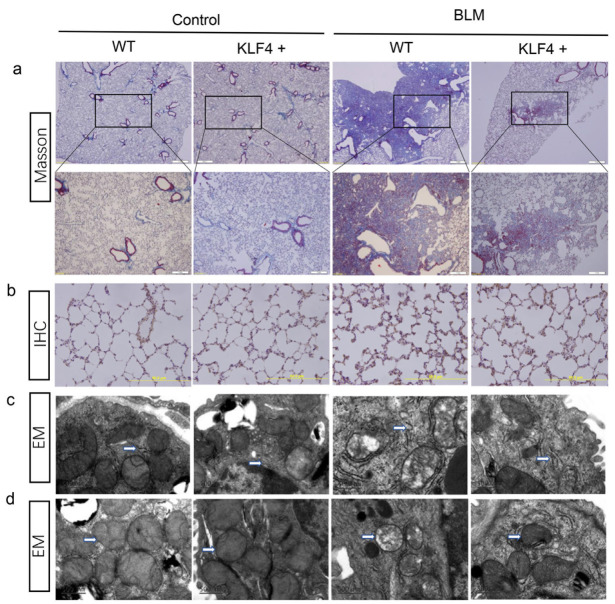
Overexpression of KLF4 inhibited bleomycin-induced pulmonary fibrosis in vivo. Mice were intratracheally administered with bleomycin to induce pulmonary fibrosis. Saline was used as a control in the KLF4-overexpressing mice group and the wild-type group. Masson staining (**a**) and collagen III immunohistochemical analysis (**b**) were used to detect collagen deposition. Representative images from six mice per group are shown. Scale bar of the Masson staining image = 1000 μm; scale bar of the immunohistochemical analysis image = 50 μm. Transmission electron microscopy was used to observe changes in the morphology of endoplasmic reticulum (**c**) and mitochondria (**d**). Representative images from three mice per group are shown. Scale bar = 200 μm.

**Figure 3 ijms-24-14008-f003:**
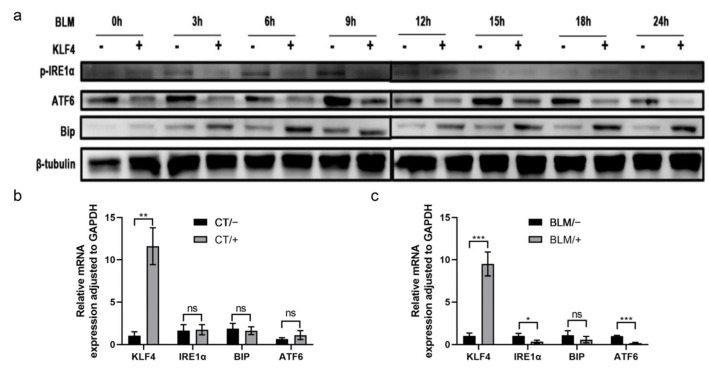
Overexpression of KLF4 attenuated ERS in bleomycin-induced fibrosis. HFL-1 fibrosis was induced by bleomycin in the KLF4-overexpressing group and the control group. Total protein was extracted at multiple time points (0, 3, 6, 9, 12, 15, 18, and 24 h). Protein immunoblotting was performed to examine the expression levels of p-IRE1α, ATF6, and BIP, with β-tubulin as a reference, as shown in (**a**). Mice were injected with bleomycin to induce pulmonary fibrosis. Tissue RNA was extracted from the KLF4-overexpressing mice group and the wild-type group treated with normal saline as a control and subjected to the RT-qPCR test to detect the mRNA expression of p-IRE1α, ATF6, and BIP (**b**,**c**). The values are expressed as mean ± SD from six samples in each group. ns implies no significance, * *p* < 0.05, ** *p* < 0.01, and *** *p* < 0.001, as determined by *t*-test.

**Figure 4 ijms-24-14008-f004:**
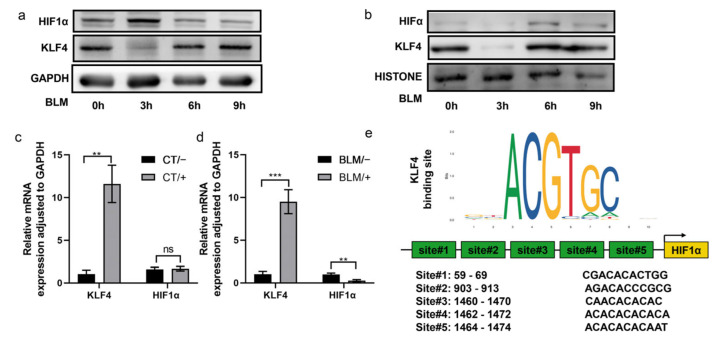
KLF4 affected the expression of HIF-1α. HFL-1 cells were treated with bleomycin to induce fibrosis. Cytoplasmic proteins (**a**) and nuclear proteins (**b**) were extracted at multiple time points (0, 3, 6, and 9 h). Western blotting assays were conducted to examine KLF4 and HIF-1α expression, with GAPDH as an internal reference. Mice were injected with bleomycin to induce pulmonary fibrosis. Tissue RNA was extracted from KLF4-overexpressing mice and wild-type mice treated with normal saline as a control for qRT-PCR test to detect the mRNA expression of HIF-1α (**c**,**d**). The values are expressed as mean ± SD from six samples in each group. ns implies no significance, ** *p* < 0.01, and *** *p* < 0.001, as determined by *t*-test. The sites of the putative KLF4 binding motif in −2000 HIF-1α promoter were predicted using the JASPAR database (**e**).

## Data Availability

All data generated or analyzed in this study are included in this paper and can be obtained from the authors upon reasonable request.

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
