# Peer review of "Overexpression of KLF4 Suppresses Pulmonary Fibrosis through the HIF-1α/Endoplasmic Reticulum Stress Signaling Pathway"

_ijms, 2023, doi:10.3390/ijms241814008_

Round 1

Reviewer 1 Report

The article is an interesting one not only for clinical purposes, but especially for clinical research in pulmonary fibrosis. The topic is original and adds value to the subject area.

The article represents a future point of view in medicine and identifying biomarkers that can be used routinely later is auspicious in medical practice.

The paper is very well written, with a strong and clear data and statistical analysis.

The conclusions are consistent with the evidence and arguments presented.

Author Response

Thank you for your recognition of our work.

Reviewer 2 Report

The topic is interesting. The manuscript is quite well written. I have some suggestions:

1) Abstract. Hypoxia-inducible Factor 1α/Endoplasmic Reticulum Stress Signaling Pathway (HIF1α/ERS) plays an important role in the pathogenesis of pulmonary fibrosis. However, the upstream regulatory mediators are still unclear. Here, through bioinformatics analysis, we found that KLF4 expression was decreased in idiopathic pulmonary fibrosis (IPF) lung tissue compared with non-IPF; in bleomycin-induced fibrotic HLF-1 cells, KLF4 expression was also decreased. In mice with bleomycin-induced pulmonary fibrosis, the fibrosis was significantly reduced in mice overexpressing KLF4 compared with wild-type mice. In the mice and HLF-1 cells, overexpression of KLF4 reduced bleomycin-induced protein expression of Hypoxia-inducible Factor 1α (HIF1α) and ERS markers, particularly p-IRE1α and ATF6. Using JASPAR database, KLF4 was predicted to have five binding sites with HIF1α promotor. Our in vitro and in vivo results suggest that KLF4 may inhibit pulmonary fibrosis through the HIF1α/ERS pathway.  Please, add the most important statistically significant values to support the results. 

2) Abstract. Our in vitro and in vivo results suggest that KLF4 may inhibit pulmonary fibrosis through the HIF1α/ERS pathway.  Please, improve the conclusions and underline the novelty of the study to make the paper more appealing and get it cited more. 

3) 1. Introduction L26-33. Idiopathic pulmonary fibrosis (IPF) is a progressive and fatal lung disease in which patients experience progressive aggravation of dyspnea and eventually respiratory failure until death[1,2]. IPF is characterized by fibroblast proliferation and extracellular matrix remodeling leading to irreversible scarring of the lung [3].  The endoplasmic reticulum is an organelle responsible for protein homeostasis, and plays an important role in folding and processing protein. Any factor that disturbs folding  and processing protein will lead to the accumulation of misfolded proteins in the endoplasmic reticulum, which is the endoplasmic reticulum stress (ERS). Please, improve support these sentences with several recent references, I suggest:

a- Evaluation of Correlations between Genetic Variants and High-Resolution Computed Tomography Patterns in Idiopathic Pulmonary Fibrosis. Diagnostics (Basel). 2021;11(5):762.  doi:10.3390/diagnostics11050762

b-Theophylline Attenuates BLM-Induced Pulmonary Fibrosis by Inhibiting Th17 Differentiation. Int. J. Mol. Sci. 202324, 1019. https://doi.org/10.3390/ijms24021019

c- RNA Sequencing of Epithelial Cell/Fibroblastic Foci Sandwich in Idiopathic Pulmonary Fibrosis: New Insights on the Signaling Pathway. Int. J. Mol. Sci. 202223, 3323. https://doi.org/10.3390/ijms23063323

d- Regeneration or Repair? The Role of Alveolar Epithelial Cells in the Pathogenesis of Idiopathic Pulmonary Fibrosis (IPF). Cells. 2022;11(13):2095. Published 2022 Jun 30. doi:10.3390/cells11132095

4)L62-65. The results we present here suggest that KLF4 can affect the progression of pulmo- nary fibrosis by regulating the HIF-1α/ERS pathway. These findings reveal that KLF4 is  an upstream regulator of HIF-1α. They suggest that drugs targeting KLF4, such as APTO- 253, in addition to phase I clinical trials in patients with advanced or metastatic solid tumors[22], may also play a role in pulmonary fibrosis.  Please insert here a complete, exhaustive description of the purpose of the study, not the results. 

6) 4.10. Statistical Analysis L283-286. All statistical analyses were carried out by SPSS software, version 24.0. Differences among sample groups were analyzed using independent t-tests or nonparametric tests. For the Western blotting examination of Collagen Ⅰand Collagen Ⅲ expression, Wilcoxon signed-rank test was used. A p < 0.05 was considered statistically significant. Please, improve this section and underline the different statistical tests used to evaluate the data. 

7) L 179-186. The study also has limitations. The specific molecular mechanism has not been ex-plored in this study. As a transcription factor, KLF4 is predicted to bind to the five sites of HIF1A through bioinformatics analysis in this study. In the next study, we will use CHIP  and EMSA to verify the combination of the two and analyze the specific binding sites point.  A small-molecule inducer of KLF4, APTO-253, has been shown to restore its expres-  sion in fibrotic fibroblasts and induce remission in an experimental model of clinically relevant persistent pulmonary fibrosis[29]. Our study provides support for the clinical application of APTO-253 in pulmonary fibrosis.  Please, improve the description of study limitations (e.g. sample size of study). Furthermore, Underline the novelty of the study and the possible clinical implications. 

Author Response

请参阅附件。

Reviewer 3 Report

I read this article with great interest. 

It's a well-written and well-designed study with could pave the way towards a better characterization of IPF. 

I have some comments:

- In the introduction section please clearly state the aim of the present study, which is missing.

- In data analysis section please specify whether sample size was calculated and how. 

- I found some minor english errors throughout the paper. Please have a deep languge revision.

- The discussion section is way too short. It should be expanded by adding possible expainations about your findings and a clear conclusion with future implications. Otherwise, this should become a brief report instead of a full lenght article. 

moderate revision 

Round 2

Reviewer 2 Report

The manuscript has been improved. No further comments

The manuscript requires only few changes regarding the English language

Reviewer 3 Report

ok to accept now.